# 11-Nor-9-Carboxy Tetrahydrocannabinol Distribution in Fluid from the Chest Cavity in Cannabis-Related Post-Mortem Cases

**DOI:** 10.3390/toxics11090740

**Published:** 2023-08-29

**Authors:** Torki A. Zughaibi, Hassan Alharbi, Adel Al-Saadi, Abdulnasser E. Alzahrani, Ahmed I. Al-Asmari

**Affiliations:** 1Department of Medical Laboratory Sciences, Faculty of Applied Medical Sciences, King Abdulaziz University, Jeddah 21589, Saudi Arabia; 2King Fahd Medical Research Center, King Abdulaziz University, Jeddah 21589, Saudi Arabia; 3Poison Control and Forensic Chemistry Center, Ministry of Health, Jeddah 21176, Saudi Arabia; 4Special Toxicological Analysis Unit, Pathology and Laboratory Medicine DPLM, King Faisal Specialist Hospital and Research Center, P.O. Box 3354, Riyadh 11211, Saudi Arabia

**Keywords:** 11-nor-Δ^9^-carboxy tetrahydrocannabinol, post-mortem analysis, fluid, chest cavity, LC-MS/MS

## Abstract

In this study, the presence of 11-nor-Δ^9^-carboxy tetrahydrocannabinol (THC-COOH) in postmortem fluid obtained from the chest cavity (FCC) of postmortem cases collected from drug-related fatalities or criminal-related deaths in Jeddah, Saudi Arabia, was investigated to evaluate its suitability for use as a complementary specimen to blood and biological specimens in cases where no bodily fluids are available or suitable for analysis. The relationships between THC-COOH concentrations in the FCC samples and age, body mass index (BMI), polydrug intoxication, manner, and cause of death were investigated. Methods: Fifteen postmortem cases of FCC were analyzed using fully validated liquid chromatography-positive-electrospray ionization tandem mass spectrometry (LC-MS/MS). Results: FCC samples were collected from 15 postmortem cases; only THC-COOH tested positive, with a median concentration of 480 ng/mL (range = 80–3010 ng/mL). THC-COOH in FCC were higher than THC-COOH in all tested specimens with exception to bile, the median ratio FCC/blood with sodium fluoride, FCC/urine, FCC/gastric content, FCC/bile, FCC/liver, FCC/kidney, FCC/brain, FCC/stomach wall, FCC/lung, and FCC/intestine tissue were 48, 2, 0.2, 6, 4, 6, 102, 11, 5 and 10-fold, respectively. Conclusion: This is the first postmortem report of THC-COOH in the FCC using cannabinoid-related analysis. The FCC samples were liquid, easy to manipulate, and extracted using the same procedure as the blood samples. The source of THC-COOH detected in FCC could be derived from the surrounding organs due to postmortem redistribution or contamination due to postmortem changes after death. THC-COOH, which is stored in adipose tissues, could be a major source of THC-COOH found in the FCC.

## 1. Introduction

Cannabis is an increasingly harmful product that is used worldwide [1]. The American National Survey on Drug Use and Health reported that 16% of the American population (43.5 million) used marijuana in 2018 [2,3]. In Canada, 3.8% of drivers were found positive for cannabis, which is reported to increase the risk of automobile accidents two-fold [4]. Cannabis is considered a substance of abuse in the Middle East and North Africa (MENA). In Arab nations, apart from Lebanon, cannabis is considered as a substance of abuse [5]. However, rarely has the reporting on the role of cannabis use in MENA investigated antemortem of postmortem cases. Indirectly, cannabis is commonly detected in postmortem cases with powerful drugs of abuse such as heroin and methamphetamine [6,7]. The impact of cannabis in the cause of death is not fully understood, and it is a paramount task for forensic toxicologists to provide information regarding its role in these deaths; therefore, forensic postmortem investigations are required. Consequently, identification of cannabinoids and their metabolites is crucial for forensic toxicologists [8,9].

In human cadavers, the active compound of cannabis (Δ^9^-tetrahydrocannabinol (THC)) metabolism is well known, and this compound has a psychoactive effect following cannabis administration [10]. THC is converted to 11-hydroxy-tetrahydrocannabinol (THC-OH) and then metabolized to 11-nor-Δ^9^-carboxy tetrahydrocannabinol (THC-COOH) [11]. Until recently, the inclusion of THC and its metabolites in postmortem analysis has gained less interest due to the common belief that it does not contribute to the cause of death. Numerous measurement methods have been reported for the analysis of these metabolites in ante-mortem specimens, particularly for drivers under the influence of drugs and workplace drug testing. Most of these challenges are encountered due to the low THC and THC-OH concentrations in the blood, and most of the THC and its metabolites are converted to more polar metabolites by glucuronidation. Free THC and its metabolites were rarely detected in urine samples. The hydrolysis step becomes a cornerstone for the detection of THC and its major metabolite, THC-COOH, in bodily and tissue specimens, regardless of the techniques used for analysis. Therefore, two challenges must be considered when developing methods for THC and metabolite analyses. First, the low concentration of active THC metabolites in the specimens of interest requires sensitive analytical techniques. Second, the samples were prepared to be suitable for testing the free analytes of interest without preliminary sample preparation for analyzing the polar conjugated metabolites of these analytes. The approach of testing glucuronide metabolites has received little attention in previous postmortem analyses for two main reasons. The first is that not all glucuronide metabolites and their internal standards are commercially available. The second reason is that testing glucuronide metabolites using gas chromatography-mass spectrometry (GC-MS) is not appropriate and is time consuming because of these polar metabolites. In postmortem testing for cannabinoids, more than one procedure is often used to obtain free THC and its metabolites in non-blood specimens; for example, a combination of protein precipitation, following by alkaline or enzymatic hydrolysis, and then subjected to either liquid-liquid extraction (LE) or solid phase extraction (SPE), followed by derivatization using GC-MS. The use of LC-MS techniques has a great impact on the postmortem testing of cannabinoids, and several advantages can be gained when using LC-MS techniques over GC-MS, such as the lack of derivatizing agents required [12,13], in cases where urine samples, direct analysis without sample pretreatment approach into LC-MS, and the most important cannabinoid glucuronide can be directly measured without hydrolysis steps [8,14].

THC and THC-COOH have been the most frequently detected cannabis metabolites in previous studies, and few studies have reported THC-OH [8,9,11,14,15]. It is well known that the distribution of parent drugs and their metabolites varies among individuals due to their tolerance, occurrence of usage, presence of other drug(s), state of health of study subjects, stability of the target analytes in antemortem and postmortem cases, and during storage in test tubes. One of the goals of testing parents and metabolites is to understand the time of intake; however, this has been poorly reported in previous studies. In fact, each cannabinoid metabolite is unique and time-dependent. THC and THC-OH appear shortly following cannabis demonstration, which is similar to THC-COOH; however, the latter could be determined in human specimens for a longer time, even without THC and THC. THC-COOH is stored in fatty adipose tissues, continues to leach into the blood circulation, and is finally excreted with urine, which makes detection of THC-COOH unsuitable for distinguishing the time of use. The appearance of THC and THC-OH in the blood has been suggested as a tool to estimate the state of impairment and recent cannabinoids used [9,11].

Blood is the sample of choice in most forensic applications, particularly postmortem analysis. A history of drug use can be evaluated using urine samples in parallel with blood tests. However, these two distinct specimens may not be available on many postmortem occasions such as traffic accidents and urination before death. Therefore, alternative specimens may be used to obtain data to support cases in hand [2,11,16]. Analysis of these alternative specimens may introduce new challenges regarding the homogeneous nature of these specimens, that is, tissue specimens (skeletal muscle, liver, gallbladder, kidneys, spleen, and brain tissues).

In situations where blood or other bodily fluids are unattainable during autopsy examination, it is crucial to explore dependable non-blood postmortem samples as alternative or complementary sources for quantifying and identifying THC and its metabolites. None of the body fluids were deemed appropriate for cases that were received at the JPCC. Alternative samples, such as liver, kidneys, stomach wall, and brain, were collected. In some cases, the fluid from the chest cavity (FCC), specifically from the pleural cavity, was sent for analysis.

Notably, high concentrations of THC-COOH were obtained when the FCC was tested for cannabinoids; this is novel, and no information regarding FCC testing is available in the literature. Thus, FCC from other cannabinoid-positive cases that conceded positive results for THC or its metabolites were further evaluated to investigate the concentration of THC-COOH in the FCC.

To the best of the authors’ knowledge, this is the first postmortem report of THC-COOH in the FCC. The impact of this approach is to provide suitable and homogenous specimens appropriate for testing analytes of interest that can be extracted easily, when compared to sold tissues specimens, which are always tested in such cases where no blood is available. It was investigated if these analytes’ concentrations differ from bodily fluids and tissue specimen’s concentration and which analytes are mostly detected in FCC. The purpose of this study was to study the value of non-blood specimens including, for the first time, FCC. In addition, these results were compared with other postmortem specimen results from previously published reports.

## 2. Materials and Methods

### 2.1. Speciemns Selection

The demography of the deceased in this report (history, sex, age, body mass index (BMI), interval time (PMI), and mode of death) was acquired from the Forensic Postmortem Jeddah database (FTRJ), which is an online database for medicolegal cases received by the Jeddah Toxicology Centre (JTC), Jeddah, Saudi Arabia.

### 2.2. Specimens Collection

In this investigation, the same protocol used for sample collection in our previous cannabinoid investigations was followed [9,15,17]. Briefly, blood was collected in sample tubes containing at least 1–2% sodium fluoride (BNaF). All BNaF was drawn from subclavian vein, while the liver was collected from at least three different sites throughout the deep right lobe of the liver in order to avoid contamination. Kidney tissues were taken from both the right and left kidneys, at the center of each kidney. Urine samples were collected from the bladder tissue using a clean syringe. Gastric contents obtained at autopsy were collected and used for analysis. Bile in liquid form was collected when it was possible. The stomach wall (W-Stomach), lung, brain, and small intestine (S. intestine) tissue were collected from 3–5 sites of these tissues.

### 2.3. Bodily and Tissues Sample Preparation

#### 2.3.1. Non-Hydrolyzed Specimens

One mL of BNaF and FCC were kept separately in 15 mL sample glass tubes (screw-capped). Next, 50 µL of internal standards containing THC-OH-d_3_, THC-COOH-d_9_, and THC-d_3_ (50 ng/mL) were transferred to each calibration and test sample and mixed thoroughly.

#### 2.3.2. Hydrolyzed Specimens

In this study, one milliliter of urine, bile, and gastric content were subjected to alkaline hydrolysis. Then, 50 µL of internal standards containing THC-OH-d_3_, THC-COOH-d_9_, and THC-d_3_ (50 ng/mL) were transferred to each calibration and test sample and mixed thoroughly. Sodium hydroxide (NaOH) (10 N) was added to each specimen (200 µL). All the urine, bile, and gastric content samples were placed in a water bath at 60 °C for at least 20 min to facilitate hydrolysis. The pH of the samples was adjusted to 3.5, using 2 mL of concentrated glacial acetic acid (GAA), and placed in a cold place for at least 5 min. For the brain, S-Intestine, W-Stomach, and Lung tissues, a gram of each tissue was weighed and then diluted using aqueous 1% sodium fluoride. Tissue at a ratio of 2:1 was added into a stomacher bag and homogenized for 5 min in a Stomacher machine (Seward Limited, Worthing, UK). Homogenized tissue (500 mg) was transferred to a new test tube. Then, 50 µL of the same internal standard was spiked into each tube. Next, 200 μL of NaOH (10 N) was transferred to each test tube, which was placed in a water bath at 60 °C for at least 20 min to facilitate hydrolysis. After cooling the sample tubes for 5 min, the sample pH was adjusted to 2 mL of GAA. The specimens were then transferred for centrifugation for 10 min at 2200× *g*. Finally, the supernatant was transferred into a clean test tube.

#### 2.3.3. Solid Phase Extraction

As detailed in the Al-Asmari report, all samples were processed using labeled Clean Screen^®^ cartridges for extraction [9,15]. Briefly, the SPE system was positioned on a vacuum manifold. Initially, the column was preconditioned by adding first adding 3 mL of methanol, then adding 3 mL of deionized water (H_2_O-D) to each cartridge, and 1 mL of hydrochloric acid (HCl, 0.1 M). Next, the supernatant of the specimens prepared above was transferred onto labelled SPE cartridges and allowed to pass through the gravity force. Next, the SPE cartridges were washed with 2 mL of H_2_O-D, followed by the addition of 1 mL of a solution containing HCl (0.1 M) and acetonitrile (70:30). The SPE cartridges were then placed under vacuum at >10 inches Hg for at least 5 min for drying. Hexane (200 µL) was added for further cleaning. After that, analytes of interest were eluted from labeled SPE cartridges by adding a mixture containing 2 mL of ethyl acetate and hexane (50:50) and then final extracts were dried using nitrogen at 40 °C. Finally, residues were mixed using 100 µL of initial percentages of mobile phase (70% methanol:30% ammonium formate solution (10 mM, pH 3.5) and 1 μL of final extract were injected into the LC-MS/MS.

### 2.4. LC-MS/MS Conditions

A previously fully validated in-house LC-MS/MS procedure to quantify and detect analytes of interest was used in this study [9]. In these reports, a triple quadrupole mass spectrometer analyzer, operated by electrospray ion, was operated using the multiple reaction monitoring positive ion mode (model: Shimadzu LCMS-8050, Kyoto, Japan) in combination with Ultra-High Performance Liquid Chromatography (UPLC, model: Nexera, Shimadzu, Kyoto, Japan). In this procedure, Raptor Biphenyl column (50 × 3.0 mm, 2.7 μm) and its guard column (model: Raptor Biphenyl, 2.7 μm, 5/3.0 mm; Restek, Bellefonte, PA, USA) were chosen for the separation of analytes; the oven column was set at 40 °C. The autosampler was maintained at 4 °C, and the flow rate was 0.3 mL/min throughout the run. The analytes of interest and their corresponding internal standards were achieved using a gradient elution containing two mobile phases: an aqueous modifier containing ammonium (10 mM, pH 3 (A)) and an organic modifier containing 100% methanol (B). The gradient mobile phase was set at 70% B for one the first min of the run, followed by an increase in B to 95% at 5 min. Solution B was maintained at 95% concentration for 3 min. Finally, 70% of the B solution was installed at 9 min and maintained for the last min of the run at 10 min.

The MS/MS setting was in accordance with our published procedure [9], and spray voltage (7000 V) and temperature (40 °C) were applied in this study. The ionized analytes were then carried into the high vacuum of the MS system. In the MRM mode, Q1 targets the m/z values at 315, 331, and 345, with consequent fragmentation of the target ions at 193, 123, 313, 20, 193, and 299 m/z for THC, THC-OH, and THC-COOH, respectively. For the Q3 fragment, daughter ions were chosen for quantitation purposes. The retention time was within 0.05 of the reference standards and the ion ratios were within +20%.

### 2.5. Method Validation

The experimental procedures were fully validated as described in detail elsewhere [9,15]. Further optimization was conducted to detect trace concentrations (TR) of THC and THC-OH. In a previous study, the lower limit of quantification (LOQ) were as low as 1 ng/mL and 1 ng/g in body and solid tissue postmortem specimens, respectively [9,15,17]. All calibration curves for THC and its metabolites were found to be linear using calibration ranges of 1–1000 ng/mL (BNaF, VH, urine, bile, gastric contents, and FCC) and 1–1000 ng/g (liver, kidney, and brain tissues) for THC-COOH. Linear calibration curves with coefficients of determination greater than 0.999 were obtained. Within-run precision and between-run precision were evaluated for THC and THC-OH using three quality control standards (QCs) of 25, 100, and 750 for bodily and solid tissues. In the current method, the within- and between-run precisions were better than 10%. The accuracy values were assessed using similar quality control as precision studies, which ranged −8% to +8%. The procedure reported by Matuszewski et al. was followed to measure the effect of the matrix and the recoveries of THC and its metabolites using QCs for each analyte [18]. The matrix effect and recoveries were acceptable, ranging from 78.0% to 122% and 79% to 97%, respectively.

As the variation between concentrations is obvious and can be dependent on health status, chronic use, weight, and other factors, dilution studies should be conducted to ensure that the method is capable of accurately detecting the analytes of interest, as dilution of specimens is also part of the method of extraction and sample preparation. The QCs samples were diluted 100 and 10 times their target concentrations and measured using the described method. All the obtained results were determined and found to be acceptable compared to their target concentrations (±15%). Method selectivity was examined by injecting commonly encountered compounds into the LC-MS/MS system to investigate their effects on the analytes of interest, and no interference with the analytes of interest was detected. The blank containing only the mobile phase was used as a sample and subjected to LC-MS/MS analysis after the injection of a high control concentration directly in order to examine any carryover effects.

### 2.6. Statistical Analysis

All statistical measurements included in this work were conducted using Statistical Packages for Software Sciences version 29 purchased from IBM Corporation (Armonk, New York, NY, USA).

## 3. Results

### 3.1. Demographic Profile

Data from the 15 FCC specimens examined in this study were evaluated. In these cases, the median age of the patients was 26 years (range: 18–50 years-old, 93% were male, and 47% were aged between 21 and 30-year-old. Patients were classified into three groups according to their BMI (normal (BMI = 18–24.9, overweight (BMI = 25–30), and obese (BMI ≧ 30)), and almost 40% of the deceased were classified as having normal BMI. The median PMIs of the current study was 48 h, and signs of putrefaction were observed in ten out of fifteen cases in the current study (three and seven of these cases were classified as partially and heavily decomposed cases). In this study, 60% and 87% of patients had a history of drug-related disorders and polydrug pharmacies, respectively (Table 1).

### 3.2. Case Samples

#### 3.2.1. THC-COOH in FCC

FCC samples were collected from 15 postmortem cases; only THC-COOH was positive (median THC-COOH = 480 ng/mL, range concentration = 80–3010 ng/mL). THC-COOH in FCC concentrations were higher than THC-COOH concentrations in all tested specimens, with the exception of bile (Table 2 and Figure 1), and the median ratios FCC/BNaF, FCC/urine, FCC/gastric content, FCC/bile, FCC/liver, FCC/kidney, FCC/brain, FCC/stomach wall, FCC/lung, and FCC/intestine tissues were 48, 2, 0.2, 6, 4, 6, 102, 11, 5, and 10-fold, respectively.

#### 3.2.2. Cause and Manner of Death

Table 3 shows that the differences between FCC and BNaF in accordance with the cause of death can be classified into three types: deaths solely due to drug overdose, death-related fatalities by the combination of drug use and other circumstances such as car accidents, falls from height, etc., and death unrelated to drugs such as violence and homicides. The THC-COOH concentration in the FCC was higher in drug-related deaths than in the drug-only and non-drug-related fatalities groups (Figure 2). No differences were observed between the THC-COOH concentration in both accidental overdose and accidental injury manner of deaths; the highest THC-COOH was determined in homicidal cases with 1160 ng/mL, and the lowest THC-COOH concentration was found in suicidal manner of deaths (190 ng/mL) (Figure 3). Figure 4 shows a higher median of THC-COOH in the FCC in heavy and partially putrefied cases than in non-petrified cases.

#### 3.2.3. History of Drug Abuse, PMIs, and BMI

The relationship between the THC-COOH concentration in the FCC and history of drug abuse showed no significant differences (Table 3). The most obvious differences were observed in the PMIs group, in which higher concentrations were observed with longer PMI times. This could be explained by postmortem redistribution from the surrounding organs. Moreover, decreasing the water content in the case of putrefaction led to the concentration of analytes in the FCC. This is supported by the high level of THC-COOH in heavily decomposed (1100 ng/mL) cases, followed by partially putrefied cases (642 ng/mL), in comparison to non-purified cases (240 ng/mL). Differences in THC-COOH in the FCC were not observed between the BMI groups in comparison with THC-COOH in the BNaF normal BMI group level than in the overweight and obese groups.

## 4. Discussion

Few studies have discussed the post-mortem data on THC, THC-OH, and THC-COOH in autopsy samples. Kemp et al. [16] reported 55 cases related to cannabis use in a series of cases of deaths related to fatal aviation accidents in the USA. This study revealed the usefulness of using alternative specimens, particularly lung tissue, in postmortem cannabinoid analysis. Saenz et al. found that the use of alternative specimens is useful when severe trauma is associated with accidents in case no blood is available. In that study, vitreous humor, brain, spleen, muscle, liver, lung, kidney, bile, heart, urine, and blood were found to be suitable alternatives or supplemental choices for qualitative cannabinoid detection [11]. Comparable conclusions were reported by Cliburn et al., in which ten pilots were involved in airplane crashes [2]. Postmortem details of cannabinoid testing in multiple body fluids and tissue analyses in forensic toxicology cases unrelated to aviation accidents have rarely been reported. Al-Asmari reported 32 cases that tested positive for cannabinoids. In that study, multiple postmortem specimens, including body and tissue samples of interest, were investigated [15]. These studies identified alternative specimens suitable for routine postmortem analysis of THC and its metabolites. These studies were limited by their small sample sizes, and no correlation was found between cannabinoids and their metabolites when multiple postmortem specimens were compared.

While blood is frequently employed as a specimen in forensic toxicology investigations, it may sometimes be unavailable owing to circumstances such as fatal accidents or putrefaction [2,8,16]. In this study, non-blood samples, especially FCC, were analyzed, and because this is the first report of this type of specimen, other bodily and tissue specimen results for the same subjects were compared with the FCC results. FCC was found to be positive at higher concentrations than most matrices analyzed in postmortem cases, except for bile, which agrees with previously reported bile concentrations in ten fatally injured pilots [2,14]. The FCC samples were liquid, easy to manipulate, and were extracted using the same procedure as the blood samples. The blood procedure was chosen because of the similar nature of the samples and the use of blood calibration curves for quantification of THC-COOH in the FCC. According to ANSI/ASB guidelines [19], it is acceptable to use a whole-blood calibration curve to quantify other postmortem bodily fluids and tissues. Although these matrices are different in terms of their viscosity, protein content, and ability to pass through SPE cartridges, a full validation of each analyte of interest in each matrix of interest may be time consuming and labor intensive, and because of the availability of samples in such unusual specimens for testing such as FCC. Another option is to adapt different calibration curves for different samples for quantification and used a control from the FACA with each batch, taking into consideration in these matrices, which is logically much higher in FCC than in blood; this is also the case with drugs and different metabolites. Therefore, a higher ULOQ of 1000 ng/mL BNaF was applied.

The importance of the FCC matrix comes from the nature of the cases, as most are traumatic, violent, or putrefied cases in which non-blood or biological fluid is available or suitable for analysis. For example, in putrefied cases, solid tissues such as the liver, lung, kidney, and brain can be tested, and their extraction is usually labor- and time-consuming compared to testing bodily fluid specimens. THC and THC-OH were rarely observed in solid tissues, and only THC-COOH was easily detected, which is similar to the result from the FCC. In Kemp et al. [16] and in other study [14], THC together with THC-COOH were determined in many specimens, including solid tissues. In that study, liver and kidney specimens showed a positive result for these analytes. In a comparable manner to this work, THC-COOH was determined in the liver and kidney, which agrees with earlier reports [11,14,15,16]. In these case study reports, THC-COOH levels ranged from 8.0 to 3894 ng/g in liver tissues and from 3 to 1774 ng/mL in kidney tissues. In contrast, the THC was negative in most cases, which agrees with the current study. The detection of THC-COOH only in most liver and kidney tissues can be explained by its role in cannabinoid metabolism and excretion from the body, as THC and its metabolites are conjugated to be eliminated in urine. THC-COOH-glucuronide degrades to THC-COOH in vivo without the need for a hydrolysis procedure during extraction, which enhances its content in these solid tissues and transfers it to surrounding bodily fluids and tissues with long PMI if tissues are not subjected to proper storage prior to death [9,15,20,21]. In a similar study on morphine conjugate metabolites, glucuronide metabolites were deconjugated to form their free forms if not immediately subjected to proper storage conditions, even without hydrolysis. This was much more obvious for analytes of interest in internal tissues, for example, in liver and kidney tissues [22]. Fluids from the chest cavity can be contaminated by postmortem interval time (PMI) and postmortem redistribution of some drugs. Postmortem changes after death could also increase the concentration of drugs, or some drugs could be degraded or evaporated [23].

In addition, the detection of free metabolites using alkaline or enzymatic procedures or instead a direct determination of conjugated metabolites of THC and its metabolite procedures has been reported, and there is always a difference between free drugs obtained using different hydrolysis methods or direct analysis [9,20]. Gronewold and Skopp [14] developed a method for the direct measurement of THC glucuronide metabolites without any hydrolysis procedure. In that study, in addition to THC metabolites, only THC-COOH-glucuronide was tested, as other glucuronide metabolites (THC-OH-glucuronide and THC-glucuronide) were not available commercially. In the report by Gronewold and Skopp, an obviously higher concentration of THC-COOH-glucuronide was found in all five cases included in their study, whereas in some cases, THC and its metabolites THC-OH were either not detected or detected at trace levels. This suggests that THC and THC-OH are significantly conjugated, which requires hydrolysis to obtain their free form, mirroring the process observed for THC-COOH. Consequently, an effective hydrolysis method is required to separate THC from THC-OH. In contrast, the free forms of these analytes were reported after alkaline hydrolysis in liver and kidney tissues, although they were present at trace levels [15,16]. Saenz et al. conducted enzymatic hydrolysis for their analysis and discovered that these analytes could be detected in liver and kidney samples [11].

As no THC or THC-OH was detected in FCC, the source of THC-COOH detected in FCC could be derived from the surrounding organs due to postmortem redistribution or contamination due to postmortem changes after death. This could also be explained by the slow transport of THC and its metabolites from the blood to organs and the lag in distribution between the blood, surrounding organs, and FCC. However, THC-COOH, which is stored in adipose tissue, could be a major source of THC-COOH found in the FCC. Other alternative specimens, such as bile, gastric contents, brain tissue, stomach wall tissues, and lung tissues, were found to be perfect matrices for the measurement of THC and THC-OH in the current study and previous investigations [2,11,15]. Al-Asmari suggested that the detection of THC in the stomach wall specimens and in the lung specimens can be justified by the influence of the route of administration as cannabinoids are smoked, thereby increasing direct contact to these tissues, thereby increasing the likelihood of THC detection in these tissues [15]. Moreover, this may be because they undergo cycles of enterohepatic movement from the liver, the site of THC, and THC-OH metabolism, in which these analytes are reabsorbed by gastric contents, which increases their concentration.

As we discuss cannabinoid use, even when using blood THC concentrations, it is still not possible to distinguish between recent and chronic use. It is believed that the concentration of cannabinoids varies depending on how often cannabinoids are consumed; however, some studies found that THC and THC-COOH could still be detected in blood up to seven days after cessation of marijuana administration [11,24]. This is because of the nature of analyte metabolism, which is deposited in adipose tissues and released into the bloodstream for a longer period when individuals do not smoke [11]. Nevertheless, in a recent study, the distribution of THC and its metabolites differed depending on renal function, body mass index, body composition, and gender [17]. Therefore, if the detection of THC can be used to confirm its recent use, this is misleading, and it is not recommended in toxicology investigations. The correlation between blood and other specimens is always poor owing to several issues such as the water content and the weight of the tissues constantly changing during the putrefying process, PMI, and postmortem redistribution phenomena, which can also affect FCC testing. However, alternative specimens can be used to provide complementary information to blood, and in cases where no blood or other traditional samples such as urine, vitreous humor, liver, and other traditionally used specimens are available at postmortem investigation [11]. Cliburn et al. believed that information obtained from these complementary specimens may add useful information to the scarce real postmortem cannabinoid case study available in the literature and advise suitable biological and tissue matrices for investigating cannabinoid-related deaths [2].

One of the limitations of using FCC is the small sample size and analysis of this specimen directly without using a hydrolysis procedure. However, in agreement with most previous investigations, poor correlations were often obtained between THC metabolite concentrations in the blood, body, and tissue specimens. This can be attributed to the lack of information regarding the route, time, and dosage of cannabinoids administered [2].

## 5. Conclusions

This is the first postmortem study to report THC-COOH in the FCC in real samples. FCC is a promising sample, as in some cases where blood and other biological fluids are unavailable, it can provide information regarding cannabinoid administration, considering that contamination from the surrounding tissues cannot be excluded. The impact of using FCC as an alternative sample is to provide suitable and homogenate specimens appropriate for testing analytes of interest that can be extracted easily compared to solid tissue specimens that are always tested in cases where no blood is available. Nevertheless, the value of the FCC matrix comes from the nature of the cases tested in this project, as most are traumatic, violent, or putrefied cases in which non-blood or biological fluid is available or suitable for analysis. The limitations of using FCC were the small sample size and direct analysis of this specimen without using a hydrolysis procedure, and more research is needed with different hydrolysis methods to investigate the presence of other cannabinoid metabolites.

## Figures and Tables

**Figure 1 toxics-11-00740-f001:**
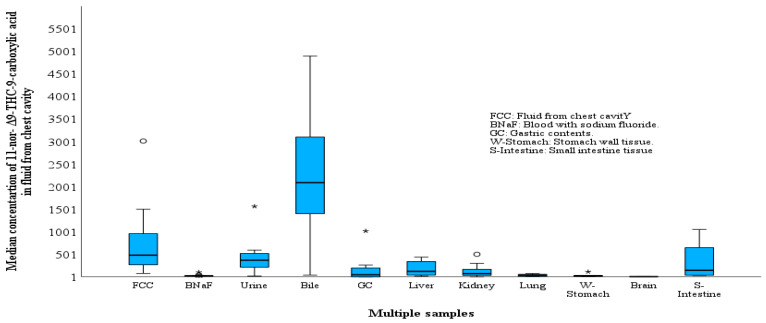
Concentrations of 11-nor-Δ^9^-carboxy tetrahydrocannabinol in different specimens, fluid obtained from the chest and/or abdominal cavities (FCC, ng/mL), blood with sodium fluoride (ng/mL), urine (ng/mL), bile (ng/mL), gastric content (ng/mL), liver tissues (ng/g), kidney tissues (ng/g), lung tissues (ng/g), stomach wall tissues (W-Stomach) (ng/g), brain tissue (ng/g) and small intestine tissue (S-Intestine) (ng/g) of tested 15 postmortem cases. The horizontal boxes represent the median concentration ratio, and the box lengths represent the 25–75th percentile. The whiskers represent the smallest and largest value within 1.5 times the interquartile range, and circles (outlier) represent values exceeding 1.5–3 times the interquartile range.

**Figure 2 toxics-11-00740-f002:**
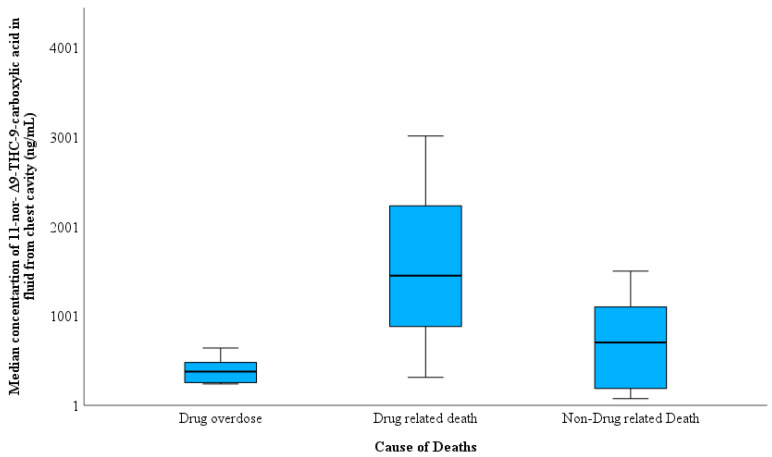
Variation in 11-nor-Δ^9^-THC-9-carboxylic acid (THC-COOH) concentration in the fluid collected from the chest cavity in 15 cases according to the cause of death.

**Figure 3 toxics-11-00740-f003:**
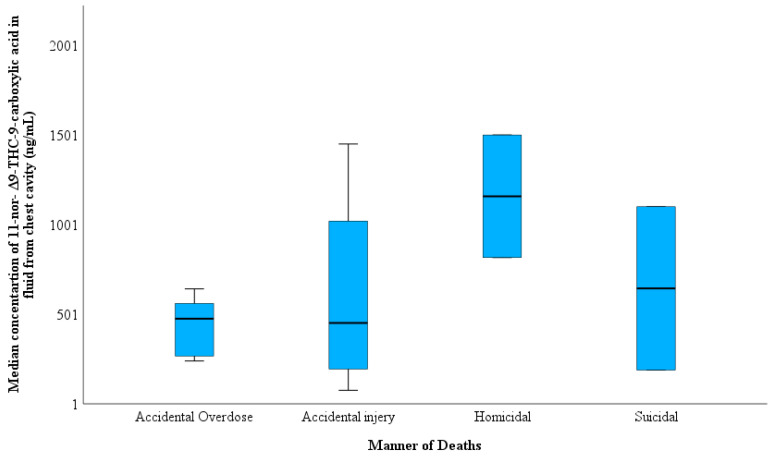
Variation in 11-nor-Δ^9^-THC-9-carboxylic acid concentration in the fluid collected from the chest cavity in the 15 cases according to the manner of death.

**Figure 4 toxics-11-00740-f004:**
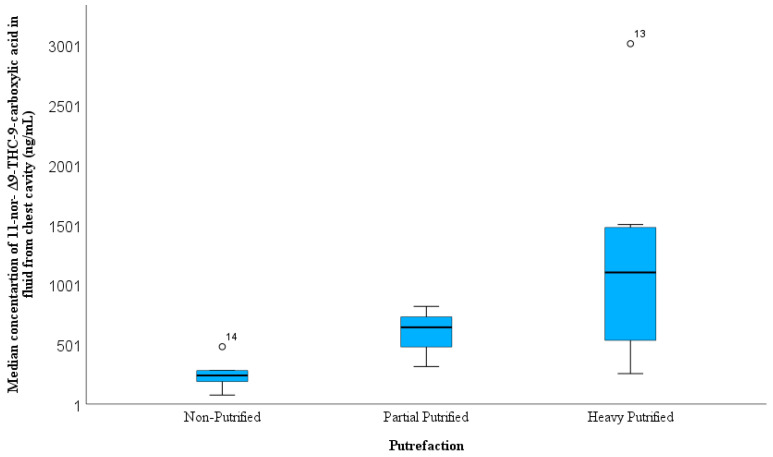
Variation in 11-nor-Δ^9^-THC-9-carboxylic acid concentration in the fluid collected from the chest cavity in 15 cases according to the degree of putrefaction.

**Table 1 toxics-11-00740-t001:** Demographic profiles of 15 patients included in this study.

Case No.	Case Information	Age	PMI	Sign of Putrefaction	Co-Ingested Substances	Location	Manner of Death	Cause of Death
1.	Deceased with history of chronic drug used was found dead in a car.	36	24	Heavily decomposed	Amphetamine	Outdoor	Accidental injury	Non-Drug related Death
2.	Fall from height, died at hospital	33	24	Non-putrefied	None	Outdoor	Accidental injury	Non-Drug related Death
3.	Poly drug intoxication, history of drug abuse	23	48	Non-putrefied	Heroin metabolites (6-monoacetylmorphine, morphine and codeine)	Indoor	Accidental overdose	Drug overdose
					Cocaine metabolites (Benzoylecgonine, ecgonine methyl ester),			
					Alprazolam			
					Pregabalin			
4.	Poly drug intoxication, history of drug abuse	23	48	Non-putrefied	Heroin metabolites (6-monoacetylmorphine, morphine and codeine)	Indoor	Accidental overdose	Drug overdose
					Ethanol			
					Alprazolam,			
					Tramadol			
5.	Gun shooting	18	24	Partially putrefied	None	Indoor	Homicidal	Non-Drug related Death
6.	Fall from height	26	72	Heavily decomposed	Methamphetamine	Outdoor	Homicidal	Non-Drug related Death
					Amphetamine			
					Alprazolam			
					Tramadol			
7.	Found dead on the street	40	96	Heavily decomposed	Amphetamine	Outdoor	Suicidal	Non-Drug related Death
8.	Found dead in his car	22	24	Heavily decomposed	Heroin metabolites (6-monoacetylmorphine, morphine and codeine)	Outdoor	Accidental overdose	Drug overdose
					Acetone			
9.	Drug overdose	21	130	Partially putrefied	Ethanol	Indoor	Accidental overdose	Drug overdose
					Barbiturates			
					Clonazepam			
10.	The deceased was found hanging	22	24	Non-putrefied	Amphetamine	Indoor	Suicidal	Non-Drug related Death
					Gabapentin			
					Alprazolam			
11.	Drug overdose	50	320	Heavily decomposed	Amphetamine	Indoor	Accidental overdose	Drug overdose
					Ethanol			
					Tramadol			
					Lidocaine			
12.	Car accident	42	72	Partially putrefied	Amphetamine	Outdoor	Accidental injury	Drug related death
13.	Found dead in his car, heavily decomposed	47	24	Heavily decomposed	Ethanol	Indoor	Accidental overdose	Drug related death
					Amphetamine			
14.	Polydrug intoxication, history of drug abuse	21	48	Non-putrefied	Heroin metabolites (6-monoacetylmorphine, morphine and codeine)	Indoor	Accidental overdose	Drug overdose
					Ethanol			
					Alprazolam			
					Tramadol			
15.	Fall from height.	37	120	Heavily decomposed	Amphetamine,	Indoor	Accidental injury	Drug related death
					Tadalafil			

**Table 2 toxics-11-00740-t002:** Summary of Δ^9^-tetrahydrocannabinol in multiple postmortem specimens from 15 cases included in this study.

		Number of Cases	Median	Minimum	Maximum
Blood with Sodium Fluoride	Δ^9^-tetrahydrocannabinol (THC)	10	10	102	20.0
	11-norΔ^9^-THC-9-carboxylic acid	11	20.0	2.0	100.0
	11-hydroxy-Δ^9^-THC	9	3.0	1.0	10.0
Fluid from Chest Cavity	THC	0	n.a. ^&^	n.a.	n.a.
	11-nor-Δ^9^-THC-9-carboxylic acid	15	480.0	80.0	3010.0
	11-hydroxy-Δ^9^-THC	0	n.a.	n.a.	n.a.
Urine	THC	4	3.0	2.0	10.0
	11-nor-Δ^9^-THC-9-carboxylic acid	10	370.0	20.0	1560.0
	11-hydroxy-Δ^9^-THC	6	2.0	Tr ^#^	4.0
Bile	THC	8	120.0	6.0	250.0
	11-nor-Δ^9^-THC-9-carboxylic acid	13	210.0	40.0	33,000.0
	11-hydroxy-Δ^9^-THC	13	110.0	3.0	1350.0
Gastric Contents	THC	10	80.0	5.0	280.0
	11-nor-Δ^9^-THC-9-carboxylic acid	9	50.0	1.0	1020.0
	11-hydroxy-Δ^9^-THC	4	4.0	1.0	30.0
Liver	THC	4	2.0	1.0	30.0
	11-nor-Δ^9^-THC-9-carboxylic acid	14	120.0	14.0	440.0
	11-hydroxy-Δ^9^-THC	2	1.0	Tr	1.0
Kidney	THC	7	2.0	1.0	70.0
	11-nor-Δ^9^-THC-9-carboxylic acid	14	70.0	1.0	500.0
	11-hydroxy-Δ^9^-THC	3	1.0	1.0	140.0
Brain	THC	3	20.0	14.0	20.0
	11-nor-Δ^9^-THC-9-carboxylic acid	3	4.0	4.0	10.0
	11-hydroxy-Δ^9^-THC	4	2.0	1.0	10.5
Stomach Wall Tissue	THC	6	30.0	4.0	470.0
	11-nor-Δ^9^-THC-9-carboxylic acid	5	30.0	6.0	110.0
	11-hydroxy-Δ^9^-THC	0	n.a.	n.a.	n.a.
Lung	THC	5	20.0	3.0	40.0
	11-nor-Δ^9^-THC-9-carboxylic acid	5	40.0	4.0	75.0
	11-hydroxy-Δ^9^-THC	1	n.a.	n.a.	n.a.
Small Intestine tissue	THC	0	n.a.	n.a.	n.a.
	11-nor-Δ^9^-THC-9-carboxylic acid	4	145.0	30.0	1050.0
	11-hydroxy-Δ^9^-THC	2	38.5	20.0	60.0

^&^ n.a: no samples, not detected or low sample size; ^#^ Tr: trace concentration.

**Table 3 toxics-11-00740-t003:** Comparison between 11-nor- Δ^9^-THC-9-carboxylic acid concentrations (ng/mL) in fluid from the chest cavity and blood with sodium fluoride according to cause of death, manner of death, location where the body was discovered, putrefaction, and history of drug abuse.

		Blood with Sodium Fluoride									Fluid from Chest Cavity		
		Δ^9^-Tetrahydrocannabinol			11-Nor-Δ^9^-THC-9-Carboxylic Acid			11-hydroxy-Δ^9^-THC			11-nor-Δ^9^-THC-9-Carboxylic Acid		
		NS *	Median	Range	NS *	Median	Range	NS *	Median	Range	NS *	Median	Range
Cause of Death	Drug overdose	2	15	10.0–19	3	24	17.0–103.0	2	5	3.0–10	6	378	240.0–1100.0
	Drug related death	3	7	2.0–10	3	18	2.0–40.0	2	1	0.5–1	3	1450	314.0–3010.0
	Non-Drug related Death	5	6	1.0–20	5	22	6.0–26.0	5	3	0.5–8	6	703	76.0–1500.0
Manner of Death	Accidental Overdose	3	10	7.0–20	4	32	17.0–103.0	3	3	1–7	7	476	240.0–3010.0
	Accidental injury	3	10	2.0–20	3	18	2.0–22.0	2	4	1–10	4	452	76.0–1450.0
	Homicidal	2	9	4.0–13	2	16	6.0–26.0	2	3	n.a	2	1158	816.2–1500.0
	Suicidal	2	3	1.2–6	2	22	n.a	2	1	n.a	2	645	n.a.
Location of death	Indoor	6	9	1.2–20	7	21	6.0–103.0	5	3	1–7	8	461	189.0–3010.0
	Outdoor	4	9	1.7–20	4	23	2.0–26.0	4	3	1–10	7	480	76.0–1500.0
Putrefaction	Non-putrefied	4	15	1.2–20	4	22	17.0–103.0	4	5	0.5–8	5	240	76.0–480.0
	Partially putrefied	2	3	1.7–4	3	6	2.0–24.0	2	2	n.a	3	642	314.0–816.2
	Heavy putrefied	4	9	6–13	4	25	18.0–40.0	3	2	0.5–3	7	1100	255.0–3010.0
Poly drug used	No	2	12	4.0–20	2	14	6.0–22.0	2	6	3.0–8	2	446	76.0–816.2
	Yes	8	9	1.2–20	9	23	2.0–103.0	7	2	0.5–7	13	480	189.0–3010.0
History of drug of abuse	No	4	9	1.2–20	4	22	6.0–26.0	4	3	1–8	6	535	76.0–1500.0
	Yes	6	9	1.7–19	7	23	2.0–103.0	5	2	1–7	9	476	240.0–3010.0

* NS: Number of samples, n.a: Not available.

## Data Availability

The data underlying this article will be shared upon reasonable request by the corresponding authors.

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
