# Peer review of "11-Nor-9-Carboxy Tetrahydrocannabinol Distribution in Fluid from the Chest Cavity in Cannabis-Related Post-Mortem Cases"

_toxics, 2023, doi:10.3390/toxics11090740_

Round 1
Reviewer 1 Report
Well designed and constructed study. Authors have taken extra efforts to find out the drug presence in the FCC. However authors should have been more clear in their explanation about the FCC that from which part of the chest cavity they collected fluid.
Author Response
We thank the author for their review. The samples were collected from the pleural cavity, and we added it to the manuscript in line 110.
Reviewer 2 Report
It is an interesting study even it is an isolated case (corpse, without adequate tissues but with chest cavity fluid). I have some major questions and some minor comments.
Major questions:
1. Why did you select stomach and intestine wall as samples? In my opinion, adipose tissue is more adequate in the case of lipophilic drugs such as THC and its metabolites.
2. In the manuscript you present different regression and correlation data. However some of them are out of the object to forensic toxicology.
- part 3.2.2, part 3.2.3 and part 3.2.5 - there is no sense to discuss the relationship between drugs and manner of death - there is no similar relationship at all when you are measuring inactive metabolite (not the THC); searching of quantitative relation between tissues which are with non-constant weight is misunderstanding. The water content and respectively the weight of the tissues are constantly changing during the putrefying process. Regression, based on only 5 points is not acceptable.
Dear authors, today we can correlate everything using software, but first, we have to keep in mind is that make sense. As a forensic toxicologist, similar correlations are just description of statistical results but without sense.
So my suggestion is to omit these parts from the manuscript.
Minor comments:
1. Please, edit the keywords: remove forensic toxicology and cannabinoids and include: 11-nor-.... and fluid chest cavity.
2. l.23 - contentS
3. l. 48 "the active component of cannabinoid"..probably cannabis instead cannabinoid?
4. l. 69, pls. edit the sentence... without preliminary sample prep...
5. l. 70 cannabinoids or cannabinoid?
6. l. 72 "flowing" ...following
7. l. 80 - THC is not cannabinoid metabolite
8. l. 109 - free THC-COOH or after hydrolysis (total)?
9. l. 129 - what is "BN"?
10. l. 129 subclavian sites or vein?
11. l. 133 "GC" - gastric content, but GC is a common abbreviation for gas-chromatography
12. l. 139 "BNaF" vs l. 129 BNF?
Professional proof editing is necessary.
Author Response
Please find our rebuttals in the document attached. Thank you for your time

Reviewer 3 Report
I read with interest the paper entitled ‘11-Nor-9-carboxy tetrahydrocannabinol distribution in fluid from the chest cavity in cannabis-related post-mortem cases.’
The work is interesting but needs some clarifications.
1) was a sample selection made or is it a case series? If so, what are the selection criteria.
2) Authors should specify whether PMI and cause of death (not mode of death as reported) played a role in the determinism of the data.
3) The authors talk about fluid in the chest cavity. What fluid is it? By what mechanism is it formed?
4) In the discussion at least hypotheses about the concentration differences found should be formulated.
5) According to the authors, can the data on the fluid of the thoracic cavity be used in any way in practice? They are certainly useful for determining the presence of the substance but hypotheses can be made on the effect of the drug in the perimortem period.
References need to be expanded.
Author Response
Please find the rebuttals in the document attached. Thank you for your time.

Round 2
Reviewer 3 Report
I have read the new version of the paper. Critical issues remain.
a) First of all, since there is no control arm, the work must be defined as a case series.
b) Secondly, the sample selection criteria are missing (the first 15 THCOOH positive cases were randomly taken)???
c) The last point should be better highlighted both in what the novelty of the study consists of and what the usefulness could be from a practical point of view. As reaffirmed by the authors, the tissue concentrations are not correlated to the perimortem blood concentrations. Is there any additional information that could be obtained in addition to understanding whether or not an individual has used cannabinoids in a more or less recent period?
Author Response
We thank the reviewer for their time, suggestions and their valuable expertise on the topic. We hope our responses are satisfactory as we responded based on our understanding of the comments. Please find the responses below:
|
a) First of all, since there is no control arm, the work must be defined as a case series. |
In this study, we presented cases with a complete analysis protocol that includes method validation and instrumentation, novel findings, promising results, and a discussion. In addition, this title describes cannabis-related postmortem cases and is in agreement with your recommendations. Therefore, no changes were made in this regard. |
|
b) Secondly, the sample selection criteria are missing (the first 15 THCOOH positive cases were randomly taken)??? |
In the introduction-Last paragraph, line 113, we mention and expand upon the following based on your comment:
"Notably, high concentrations of THC-COOH were obtained when the FCC was tested for cannabinoids; this is novel, and no information regarding FCC testing is available in the literature. Thus, FCC from other cannabinoid-positive cases that conceded positive results for THC or its metabolites were further evaluated to investigate the concentration of THC-COOH in the FCC. To the best of our knowledge, this is the first postmortem report of THC-COOH in the FCC."
|
|
c) The last point should be better highlighted both in what the novelty of the study consists of and what the usefulness could be from a practical point of view. As reaffirmed by the authors, the tissue concentrations are not correlated to the perimortem blood concentrations. Is there any additional information that could be obtained in addition to understanding whether or not an individual has used cannabinoids in a more or less recent period? |
We added the and expanded upon the following paragraphs based on your suggestion: In the introduction: last paragraph-line 117-124 “To the best of the authors’ knowledge, this is the first postmortem report of THC-COOH in the FCC. The impact of this approach is to provide suitable and homogenous specimens appropriate for testing analytes of interest which can be extracted easily, compared to sold tissues specimens, that are always tested in such cases where no blood is available. It was investigated if these analytes’ concentrations differ from bodily fluids and tissue specimen’s concentration and which analytes are mostly detected in FCC. The purpose of this study was to study the value of non-blood specimens including for the first time FCC; in addition, these results were compared with other postmortem specimen results from previously published reports.” In the discussion part line 344 The importance of the FCC matrix comes from the nature of the cases Based on your recommendation we added the following paragraph in line 389: “As we discuss cannabinoid use, even when using blood THC concentrations, it is still not possible to distinguish between recent and chronic use. It is believed that the concentration of cannabinoids varies depending on how often cannabinoids are consumed; however, some studies found that THC and THC-COOH could still be detected in blood up to 7 days after cessation of marijuana administration [11,24]. This is because of the nature of analyte metabolism, which is deposited in adipose tissues and released into the bloodstream for a longer period when individuals do not smoke [11]. Nevertheless, in a recent study, the distribution of THC and its metabolites differed depending on renal function, body mass index, body composition, and gender [17]. Therefore, if the detection of THC can be used to confirm its recent use, this is misleading, and it is not recommended in toxicology investigations. The correlation between blood and other specimens is always poor owing to many issues such as the water content and the weight of the tissues constantly changing during the putrefying process, PMI, and postmortem redistribution phenomena, which can also affect FCC testing. However, alternative specimens can be used to provide complementary information to blood, and in cases where no blood or other traditional samples such as urine, vitreous humor, liver, and other traditionally used specimens are available at postmortem investigation [11]. Cliburn et al believed that information obtained from these complementary specimens may add useful information to the scarce real postmortem cannabinoid case study available in the literature and advise suitable biological and tissue matrices for investigating cannabinoid-related deaths [2].”
|